# Green and Chemical Silver Nanoparticles and Pomegranate Formulations to Heal Infected Wounds in Diabetic Rats

**DOI:** 10.3390/antibiotics10111343

**Published:** 2021-11-03

**Authors:** Renan Aparecido Fernandes Scappaticci, Andresa Aparecida Berretta, Elina Cassia Torres, Andrei Felipe Moreira Buszinski, Gabriela Lopes Fernandes, Thaila Fernanda dos Reis, Francisco Nunes de Souza-Neto, Luiz Fernando Gorup, Emerson Rodrigues de Camargo, Debora Barros Barbosa

**Affiliations:** 1Department of Dental Materials and Prosthodontics, School of Dentistry, São Paulo State University (UNESP), Araçatuba 16015-050, SP, Brazil; renanfernandes_91@hotmail.com (R.A.F.S.); fernandesgabriela@hotmail.com (G.L.F.); thailaf@hotmail.com (T.F.d.R.); 2Laboratory of Research, Development & Innovation, Apis Flora Industrial e Comercial Ltda., Ribeirão Preto 14020-670, SP, Brazil; andresa.berretta@apisflora.com.br (A.A.B.); elinacassia@hotmail.com (E.C.T.); andrei.buszinski@apisflora.com.br (A.F.M.B.); 3Department of Chemistry, Federal University of São Carlos, São Carlos 13565-905, SP, Brazil; francisco_nsn@yahoo.com.br (F.N.d.S.-N.); lfgorup@gmail.com (L.F.G.); camargo@ufscar.br (E.R.d.C.)

**Keywords:** silver, nanoparticles, *Punica granatum*, wound healing, *Staphylococcus aureus*, *Candida albicans*

## Abstract

Infected cutaneous ulcers from diabetic rats with *Candida albicans* and *Streptococcus aureus* were treated with spray formulations containing green silver nanoparticles (GS), chemical silver nanoparticles (CS), or pomegranate peel extract (PS). After wound development and infection, the treatments were performed twice per day for 14 days. The wound healing was analyzed on days 2, 7, and 14 through the determination of CFUs, inflammatory infiltrate, angiogenesis, fibroplasia, myeloperoxidase, and collagen determination. Expressive improvement in wound healing was noted using both silver nanoparticles for 7 days. All the treatments were superior to controls and promoted significant *S. aureus* reduction after 14 days. CS presented better anti-inflammatory results, and GS and CS the highest number of fibroblasts. Despite the techniques’ limitations, GS and CS demonstrated considerable potential for managing infected wounds, especially considering no early strategies prior to the drugs, such as the debridement of these wounds, were included.

## 1. Introduction

The skin is the largest human organ that plays an important role in human homeostasis, acting as a physical barrier against pathogen invasions [1]. Eventually, the skin can suffer injuries called wounds, representing the loss of skin integrity, thereby disrupting its cellular and anatomic structures, and altering its functionality [1,2]. Once skin damage occurs, a complex response is activated, culminating with the wound-healing process [3]. This is a dynamic process that leads to skin repair through a cascade of events whose main objective is to re-establish the integrity of the damaged area [4]. This process basically encompasses the following three overlapping phases: inflammation, cellular proliferation, and remodeling [5,6]. Cutaneous wound healing quickly results in a scar formation [7]. However, several pathological conditions may lead to clinical complications that interfere with the healing success [8]; such patients at risk are diabetic people. Diabetes is a chronic disease where wound healing is a weakened process requiring drug administration and sometimes wound surgery or excision [9]. If the wound healing event does not accordingly progress, a chronic wound arises, which represents a clinical complication, increasing risks to the patient and treatment costs. Thus, for diabetic people, chronic and persistent wounds are a very impacting disease and show high rates of mortality [10,11]. Among chronic diseases, diabetes is a growing group, and it is estimated that, by 2030, about 350 million people globally will be affected by this disease [12,13].

Biofilms are very organized and dynamic communities of different microorganism species embedded in an extracellular matrix [14]. In addition to microorganisms, biofilms are composed of polysaccharides, proteins, nucleic acids, and extracellular DNA molecules [15,16]. Biofilm chemical composition, combined with its structure, makes the effectiveness of chemical therapies harder, related to increased resistance to antimicrobial drugs, hindering the action of immune cells, and can further promote chronic wound formation, increasing the risk of systemic infections [16,17]. Biofilm establishment directly interferes, hampering wound healing [18]. Biofilms are responsible for the development and maintenance of about 2% of the American population presenting chronic wounds. The lack of proper care can lead to amputations [12], with more incidents in this situation occurring in diabetic groups (about 24%) [12,19].

Recently, many efforts have been made to develop new agents targeting the healing process that avoids amputation and other complications related to chronic wounds. The search for efficient drugs against microorganisms in both the planktonic state and biofilm structures, besides presenting anti-inflammatory effects, is an open field that can be widely explored [17]. In this sense, there is interest in the use of nanotechnology and phytotherapy focusing on biofilms and ulcer treatment [20,21].

The use of natural extracts applied as antimicrobial drugs is a growing field in science that has demonstrated interesting and promising results [22,23]. Following this scenario, several plant extracts were studied for the treatment of wound healing, such as extracts from *Chrozophora tinctoria*, *Hibiscus rosa-sinensis* L., *Pongamia pinnata*, *Persea americana*, *Chamaemelum nobile* L., and *Punica granatum* [24,25,26,27,28,29,30].

Additionally, the use of silver nanoparticles (AgNPs) is widely studied and applied, such as with incorporation in shampoos, soaps, detergents, cosmetics, medical devices, and other applications. As with other nanometric materials, AgNPs are ordinarily defined as presenting at least one special dimension in the size range of 1–100 nm [31]. Although shape and size are the easiest method to classify a system, the most important property is related to its energy [32]. Nanoscience is the manipulation of a sufficiently small number of atoms to form particles with energetic characteristics that are intermediate between a single atom and a macroscopic sample. This is the reason why chemists and material scientists develop new techniques to control the number of atoms in their nanoparticles. If a molecule is a fixed number of atoms organized in a precise structure, a nanoparticle can be prepared with a different size and shape to tune some specific property of interest [32]. In the last decade, AgNPs have been intensively investigated due to their unique physical, chemical, and biological characteristics [33]. Indeed, in view of the potential application of AgNPs for sundry purposes in different fields such as electronics, catalysis, and optics, in a broad range of consumer items, and in several biomedical applications, especially due to their broad-spectrum antimicrobial activity [20], the increase in AgNP production per year, according to Rónavári et al. (2017) [34], may, globally, be near to hundreds of tons.

The use of silver nanoparticles as antimicrobial agents is increasing in different areas such as in the cosmetic, medical device, and textile industries [35,36,37,38]. For instance, AgNP-based wound dressings have been commercially available for over two decades, e.g., Acticoat™ (Smith and Nephew, London, UK), to treat a range of wounds in the clinic, including burns [39], toxic epidermal necrolysis [40], Steven–Johnson syndrome and pemphigus [41], and chronic ulcers [42].

Despite the wide application, conventional silver nanoparticles produced through the chemical route, in addition to its high costs, also show relevant side effects, such as high levels of toxicity, to mammalian cells [43]. These situations limit the application of products based on this type of nanoparticles produced through the conventional methods. On the other hand, the addition of natural compounds (e.g., plant extracts) along with silver ions during silver nanoparticle biosynthesis leads to a reduction in silver ions by those natural compounds, satisfactorily decreasing final toxicity [44] in a route named green synthesis.

Recently, our group synthesized and evaluated the cytotoxic effects and antimicrobial activity of novel spray formulations based on silver nanoparticles, comparing nanoparticles synthesized through the green or conventional routes [45]. Our results demonstrated that the formulation containing green nanoparticles (i.e., prepared with pomegranate peel extract) had better antimicrobial activity against *S. aureus* and *C. albicans*. Additionally, this green formulation presented reduced levels of cytotoxicity when compared to the spray formulation containing silver nanoparticles produced through the chemical route [45]. In the current study, we proposed evaluating the effects of spray formulations of silver nanoparticles synthesized by green and chemical routes, and pomegranate peel extract spray on the healing of infected wounds produced in an in vivo murine mode. Our results suggest that both the chemical and green syntheses of silver nanoparticles were effective for treatment of the Wistar rat model of chronic wound. Although these different formulations have shown potential application for the treatment of wounds, the wound healing process takes place through different mechanisms of action. This study was developed with researchers from multidisciplinary areas, with the mutual goal of developing a spray containing silver nanoparticles synthesized through pomegranate peel extract to treat infected wounds in diabetic patients and people with a pressure ulcer. The choice of using pomegranate peel extract is due to its great pharmacological properties including anti-inflammatory, antioxidant, and antimicrobial potential. The fact that the present study was developed in partnership with the industry gave us a very optimistic perspective regarding the commercialization of our idea.

## 2. Results

### 2.1. Wound Healing Activity

In a previous study, we demonstrated that spray formulations composed by AgNPs synthesized through a green route using pomegranate peel extract, under in vitro conditions, showed higher antimicrobial activity against *S. aureus* and *C. albicans* when compared to AgNPs synthesized through a conventional chemical route [45]. In addition, the green product presented reduced levels of cytotoxicity when compared to the spray formulation produced though the chemical route [45]. Aiming to characterize the in vivo effects of previously developed formulations, we evaluated the wound healing activity of the chemical spray (CS), green spray (GS), and pomegranate spray (PS) formulations in a model of diabetic rats. As control groups, untreated animals (C) or animals treated with silver sulfadiazine (Sulf) were included. Our results demonstrated that treatments performed with silver nanoparticles obtained by the green and chemical routes, GS and CS, respectively, presented significant wound healing activity after 7 days of treatment (Figure 1A,B). Furthermore, treatment with PS showed a significant increase in wound healing after 14 days of treatment compared to that of the other treatments. Overall, our results suggest that the silver nanoparticles (GS and CS) and pomegranate spray (PS) showed better results than the control group (C) and the sulfadiazine treatment (Sulf).

### 2.2. CFU Determination from Ulcers

To obtain more insights about the antimicrobial activity exerted by the target analyzed formulations here, the number of CFUs was directly assayed from animal ulcers after 2, 7, and 14 days after each specific treatment. Although the CS, GS, and PS formulations had shown considerable wound healing activity, no CFU reduction was observed for C. albicans either after treatment with those three formulations or for the C and Sulf groups (Figure 2A). After 14 days of treatment, the wounds infected with S. aureus presented a reduction in CFUs when treated with CS, GS, PS, or Sulf (Figure 2B). Taken together, the results suggest that all the evaluated formulations were effective against the pathogenic Gram-positive bacterium but had no influence on reducing viable C. albicans cells.

### 2.3. Quantitative Image Evaluation for Inflammatory Infiltrate, Angiogenesis, and Fibroplasia

To further characterize the inflammatory infiltrate promoted by the target formulations (CS, GS, PS, and Sulf), we quantified the inflammatory infiltrate using the histological approach (HE) followed by image quantification after 2, 7, and 14 days of treatment (Figure 3). The SQ formulation reduced the inflammatory infiltrate after 7 days (Figure 3B), while no reduction was observed at that point for the CS or Sulf treatments (Figure 3B). PS administration triggered an increase in inflammatory infiltrate after 2 and 7 days of treatment, with this profile being statistically significant only after 7 days of treatment, suggesting that wound healing induced by this product was a consequence of a proinflammatory response. Overall, after 14 days of treatment, the inflammatory response was similar for all the analyzed groups (Figure 3B).

The triggering of fibroplasia is a hallmark of wound healing progression. Due to this, we evaluated the fibroplasia levels induced by the application of spray formulations CS, GS, PS, and Sulf (Figure 3C). After 2 days of treatment, GS and CS induced a greater number of fibroblasts when compared to the other treatments. This increased number of fibroblasts may have been responsible for the better wound closure for these two groups after 7 days, as observed in Figure 1. After this period, none of the treatments showed a statistically significant difference in terms of fibroplasia induction.

Wound healing in diabetic individuals could be improved by the treatments that promote an increase in blood vessels. Here, we investigated if the pulverization of CS, GS, and PS formulations, and Sulf would be important to increase the number of blood vessels (Figure 4). According to our experiments, the spray formulations were not able to increase blood vessels in diabetic rats (Figure 4A,B).

### 2.4. Dosage of Enzyme Myeloperoxidase (MPO)

MPO is an enzyme that belongs to the heme peroxidase superfamily. This enzyme is produced by neutrophils and is biochemically used to measure the inflammatory process. Aiming to evaluate the inflammatory process triggered by GS, CS, PS, and Sulf, we measured the production of MPO from ulcer lesions recovered from the animal groups treated with these formulations. After 2 days of treatment, none of the formulations presented differences in terms of MPO production when compared to the control group (Figure 5). After 7 days of treatment, animals from the PS group maintained the same MPO level observed after 2 days, while all the other treatments, including the C group, presented reduced levels of MPO (Figure 5). These results suggested that the PS formulation could stimulate wound closure through a proinflammatory process (Figure 5). This finding corroborates the previous result observed in histological analysis (Figure 2A). Lastly, after 14 days of treatment, all the groups demonstrated an absence of inflammatory response, and no MPO was detected (Figure 5).

### 2.5. Collagen Evaluation

Collagen production is an important event that happens during the wound healing process and skin formation. Due to this, we used an indirect method to assess the collagen deposition through a hydroxyproline measurement, a derivative of proline in the collagen production pathway. After 2 days, treatments with CS and GS sprays increased the levels of hydroxyproline (Figure 6), corroborating the increased number of fibroblasts promoted by these two treatments (Figure 3C). In the later time points, no differences were observed between the treatments. These results suggest that both CS and GS induce collagen production and are probably related to the better closure shown in Figure 1. After 2 days of treatment, all the groups showed similar patterns of collagen production.

## 3. Discussion

The emergence of multidrug-resistant microorganisms is a global concern that is related to the indiscriminate use of commercially available antimicrobial drugs. In addition, the increasing number of patients with chronic diseases, such as arterial hypertension and diabetes, drives the search for new biomaterials that may be used as medicinal drugs. Therefore, considering the increase in the global diabetic population added to the rise in antimicrobial resistance, the current study focused on the characterization of spray formulations containing silver nanoparticles, produced by both chemical and green routes, and a pomegranate peel extract spray. The spray formulations were evaluated in terms of their ability to improve wound healing in diabetic rats with manually infected cutaneous ulcers. Additionally, their effects on fibroplasia, collagen deposition, angiogenesis, inflammatory effects, and antimicrobial activity were evaluated from ex vivo samples. All the treatments were compared with the control (without treatment) and silver sulphadiazine (as a reference standard treatment).

The healing process is linked to several biological events that culminate with total wound closure and healing. During this process, re-epithelialization, fibroplasia, angiogenesis, and extracellular matrix production are important steps that are generally dependent on the immunological condition of the host [23,46,47,48]. The use of natural compounds represents an alternative to chemical compounds used to induce healing, and it showed satisfactory results [23,24,26,49]; the use of pomegranate extracts to accelerate the healing process was described [1,22,25,50]. Additionally, silver nanoparticles (AgNPs) have drawn attention due their versatile functions such as antimicrobial, anti-inflammatory, and antiangiogenetic properties, and wound healing promotion, and they are used as interesting tools to overcome emerging medical challenges [51]. Besides the advantageous use, AgNPs can be a high-cost process and are often considered to be a harmful approach due the use of chemical compounds and the production of their byproducts [52,53]. On the other hand, the use of plants for silver nanoparticle synthesis was extensively explored [52,54,55]. In this case, bioactive plant compounds are used to reduce silver, avoiding the use of chemical reducers.

Here, we demonstrated that the pulverization of chemical or green silver nanoparticles improved the wound healing process in diabetic rats mainly after 7 days treatment. The formulations containing silver nanoparticles (SC and SG) presented the best results when compared to the pomegranate spray formulation without nanotechnology (SP). This can be explained by the fact that AgNPs are efficient in promoting the differentiation and maturation of keratinocytes through the stimulation of skin stem cells [56]. AgNPs can also stimulate cell maturation, promoting the differentiation of fibroblasts into myofibroblasts during healing [56]. Our data show for the first time that the use of pomegranate silver nanoparticles in a spray formulation can improve the healing process in an animal model of diabetic rats. Ragab and colleagues (2019) evaluated the effects of a hydrogel formulation prepared from silver nanoforms of *P. granatum* peel extract that demonstrated efficacy in the healing of chronic wounds in rats [1], corroborating our results. The application of an ointment based on *Prosopis juliflora* leaf extract and silver nanoparticles (PJL–AgNPs), along with Carbopol, promoted wound contraction and accelerated the healing process in mice in comparison to the negative and positive control groups [52].

The application of the PS formulation without nanotechnology was able to promote the healing process mainly after 14 days of treatment. Corroborating this, a study demonstrated that the use of an ointment containing 15% pomegranate performed about 93% of wound contraction in rats after 16 days of treatment [22]. In addition, the topical application of a product prepared with standardized pomegranate extract powder demonstrated that the formulation containing 10% peel extract was more efficient in terms of promoting the re-epithelialization of skin burns in rats [50]. Yuniarti and colleagues (2018) [57] observed that, after 14 days treatment, the use of 7.5% standardized pomegranate extract accelerated the healing process of incised wounds in the animals. Additionally, the use of a formulation with 5% extract-based ointment improved the healing process in guinea pigs when compared to the placebo group, although the results obtained from the group treated with commercial products were slightly better [25]. Overall, the formulations used in these studies had higher concentrations of active ingredients than ours did. In this case, although all the aforementioned studies that analyzed the use of pomegranate extract in promoting the healing process, the used wound models and animal species were different, which could interfere in this comparison. Taken together, these results emphasize the advantageous use of pomegranate products applied to the healing of chronic wounds.

The antimicrobial and anti-inflammatory properties of pomegranate are due to the presence of polyphenols (e.g., ellagic and gallic acid, and tannins), in addition to other benefits such as its antioxidant and antiallergic effects [30,58,59,60]. The antimicrobial activity of our formulations was assayed in ex vivo samples from rat skin ulcers manually infected with *C. albicans* or *S. aureus*. At a later time point (14 days of treatment), the CS, GS, and PS formulations, and sulfadiazine reduced the number of CFU of wounds infected with *S. aureus,* suggesting that all these products played antibiotic functions in the evaluated model of skin ulcers. Adibhesami and colleagues (2017) [61] obtained similar results concerning the anti-*S. aureus* activity of silver nanoparticles applied to the healing process in mice. On the other hand, CS, GS, and PS were not able to statistically reduce the number of CFU of *C. albicans;* a very similar profile was observed in the group of animals treated with sulfadiazine, suggesting that none of these formulations had anti-*Candida* activity. Although the results of the antibiofilm action were not so expressive for *C. albicans*, a study by Paosen and colleagues (2019) [62] reported that green AgNPs increased the membrane permeability of *S. aureus* and *C. albicans* and altered the morphological microbial cells, especially shrinking deformation in regard to *C. albicans.* Additionally, factors such as interkingdom interactions must be taken into account since, as reported by Kean et al. (2017), the biofilm formation of *C. albicans* and *S. aureus* favored strengthening both, increasing resistance to fluconazole [63]. The resistance behavior of yeasts could also be attributed to the possible production of quorum-sensing molecules by *C. albicans* such as farnesol, a molecule that regulates several functions in the biofilm, including growth control [64].

Current strategies for wound management include first cleansing followed by wound debridement [65]. Even without a thin film formed by bacteria and fungi being visible on the wound surface, a biofilm is present [65] and should be appropriately dressed. In our work, nondebridement did not permit the structural disorganization of the biofilm. Therefore, protective biofilm factors such as the uninterrupted production of the extracellular matrix and quorum-sensing molecules keep it well-structured and with active infection, even under appropriate dressings. For further studies involving infected wounds in animal models, biofilm management through antibiotics and the early strategies proposed by Murphy et al. (2020) help the healing process on both acute and postoperative wounds [65].

The induction of fibroplasia is an important effect triggered by many compounds that induce the healing process, and it is linked to tissue reparation. Our results suggest that AgNP GS and CS are not cytotoxic and induce a substantial increase in the number of fibroblasts, specifically after 2 days of treatment, further improving the results for CS nanoparticles. Furthermore, at this time point, the indirect method used here to assess collagen deposition, hydroxyproline measurement, demonstrated that these two groups of animals also presented increased levels of this amino acid. This result suggests that, in addition to fibroplasia, collagen deposition was increased in these groups at initial time points. Ragab and colleagues (2019) demonstrated under in vitro conditions that *P. granatum* AgNPs (specifically AgNP PGPEA and AgNP PGPC) had low cytotoxicity against a human normal cell line (HFB4) after encapsulation into hydrogels, once more elucidating that the use of AgNPs can offer benefits without being cytotoxic [1].

Aslam and colleagues (2006) [66] reported that the addition of pomegranate peel extract ranging from 0.005 to 0.5 µg/mL had stimulated fibroblast proliferation and type I procollagen synthesis under in vitro conditions. In terms of collagen deposition, the use of a formulation containing 10% peel extract promoted the production of high-density collagen deposition followed by low levels of proinflammatory cells [50]. In addition, the group of animals treated with sulfadiazine had no increase in fibroblasts, and showed neither an increase in the number of fibroblasts nor an increase in hydroxyproline. The use of silver sulfadiazine does not, in fact, induce collagen formation, leading to the delay of both wound closure and re-epithelization [50,67]. In summary, our results related to the improvement of healing, fibroblast proliferation, and hydroxyproline levels are in accordance with the expected tissue recovery process. Furthermore, these results are supported by Adibhesami et al. (2017), who proposed the capacity of silver nanoparticles in reducing the required time for hyperactive cells (myofibroblasts) responsible for wound contraction.

When an injury happens, the inflammation process helps in terms of the removal of damaged cells and the promotion of vasoconstriction [57]. Additionally, the recruitment of immune cells such as neutrophils and macrophages leads to the production of proinflammatory mediators (e.g., cytokines and interleukins). The inflammatory response generally happens under an equilibrated scenario once the prolonged inflammatory events can be turned into a chronic situation, which is harmful for the wound healing process. Ag has anti-inflammatory properties in addition to increasing re-epithelization [68], and silver nanoparticles may improve anti-inflammatory activity in wounds, accelerating this repair process [56]. The anti-inflammatory effects of pomegranate extracts were also widely explored [50,69,70], and punicalagin is highly involved with this anti-inflammatory capacity [69,70].

We quantified the inflammatory infiltrate of the animal groups treated with our formulations, and after 7 days of treatment, the recruitment of proinflammatory cells was reduced in all the groups, including SG and SC, with the more expressive reduction seen in the SC group. This finding corroborates the previous result obtained for the induction of fibroplasia, implying that in these groups, the pomegranate formulations reduced the inflammatory response process after 7 days treatment. Lastly, after 14 days of treatment, all the groups presented very low levels of inflammatory infiltrate. A recent study explored the anti-inflammatory response triggered by garlic silver nanoparticles (G–AgNPs) through an in vitro model measuring the denaturation of protein (BSA) as an indicator of the inflammatory process [71]. The authors demonstrated that low doses of G–AgNPs (10 μM) displayed anti-inflammatory effects comparable to those of diclofenac sodium (standard drug).

The use of products based on pomegranate extracts has been evaluated in different animal models and also in cell cultures. The application of an ointment containing 10% pomegranate peel extract reduced the inflammatory process in the burn wounds of rats after 15 days [50]. The addition of 10.0 µg/mL pomegranate peel extract was also evaluated on bovine mammary epithelial cells and decreased the mRNA accumulation of important proinflammatory genes such as TNF, IL1B, and IL10 by 18.0, 25.7, and 27.5%, respectively [69]. Surprisingly, in addition to the results for inflammatory infiltrate, an increase in MPO levels was observed in the group of animals treated with PS, suggesting that this treatment induced a proinflammatory instead of anti-inflammatory condition. This proinflammatory activity profile can be grounded in the literature, where some studies show the production of proinflammatory cytokines [72,73,74]. These studies reported that pomegranate inhibits the p38-mitogen-activated protein kinase (p38- MAPK) pathway and transcription factor NFkB (nuclear factor kappa-light-chain-enhancer of activated B cells). Thus, when p38-MAPK and NF-kB are activated, there is an increase in the expression levels of COX-2, TNF-α, MCP1, IL-1β, and iNOS, which are inflammatory mediators [74].

## 4. Materials and Methods

### 4.1. Spray Formulations

The spray formulations obtained by Fernandes and colleagues (2018) were used in the current work. Briefly, after obtaining the peel pomegranate extract and the synthesis of silver nanoparticles by both the Turkvich methodology [45,75] and green route [45], the spray formulations were prepared, taking into account the MIC values of each active ingredient and considering previous results of cellular toxicity [45]. The formulation without actives contained 0.1% carboxymethylcellulose (Labsynth, Diadema, Brazil), 7% propylene glycol (Labsynth, Diadema, Brazil), and 0.1% methylparaben (Labsynth, Diadema, Brazil). The concentrations of the active ingredients were 94 µg/mL *Punica granatum* peel extract, and 337.5 µg/mL green silver nanoparticles or 5.55 µg/mL chemical nanoparticles.

### 4.2. Ethics Statement

All protocols adopted in this study encompassing animals were approved by the local Ethics Committee for Animals from the Faculty of Medicine of Ribeirão Preto, University of São Paulo, Brazil, (process nº 140/2016)

### 4.3. In Vivo Models for Wound Healing Activity

Male Wistar rats (180–200 g) were obtained from the Central Bioterium of the Medical School of Ribeirão Preto, University of São Paulo, Brazil. The animals were kept in isolation conditions in individual cages, with water and feed ad libitum, and alternating cycles of luminosity every 12 h. Animals received streptozotocin (50 mg/kg) intravenously via the caudal artery. After 24 h, blood glucose concentration was checked for diabetic induction [76].

After confirmation of the induction of diabetes, the rats were weighed and anesthetized intraperitoneally with 70 mg/kg ketamine and 12mg/kg xylazine. Later, the rat dorsum was trichotomized and two surgical excisions were performed with a punch of 1.5-centimeter diameter, thus leaving the dermoepidermal region affected. After this procedure, the microorganisms were adjusted to 104 CFU/mL for *Staphylococcus aureus* and *Candida albicans*, then directly pipetted (100 μL) onto the lesion. The ulcerated skin was collected to be used as control (Day 0) nonulcerated skin. After surgery, oral dipyrone, 50 mg/kg body weight, diluted in saline solution every 12 h for the first 48 h was orally administered according to the behavioral changes of the animals regarding pain.

The animals were divided into 5 groups containing 6 animals each, and analysis was performed at 2, 7, and 14 days after treatments. Each animal was topically treated twice a day, at the beginning of the morning and at the end of the afternoon. The groups were (C) animals that received no treatment; (GS) green spray formulation containing green silver nanoparticles, (CS) chemical spray formulation with chemical silver nanoparticles, (PS) pomegranate spray, and (Sulf) commercial silver sulfadiazine.

### 4.4. Wound Healing Activity

At the end of the assay, animals were euthanized in a CO2 chamber after 2, 7, and 14 days. The lesions were photographed with a Sony DSC-P100 digital camera coupled to a base containing a millimeter ruler, with a standard distance of 30 cm, perpendicular to the ulcers, and the lesion areas were measured with Image J software. Then, the ulcer healing index was determined, which is equivalent to the quotient of the difference between initial and final areas divided by the initial area [77].

### 4.5. Material Collection for Future Studies

After photo acquisition, the animals had all skin trimmed around the ulcers and through the 1.5-centimeter diameter punch. The recovered materials were collected and packaged in separated tubes. Separated samples were used for the following purposes:

Sample 1: tissue was packed into an Erlenmeyer flask containing 4% formaldehyde in Sorensen phosphate buffer 0.1 M, pH 7.3 for histological study (inflammatory infiltrate, angiogenesis, and fibroplasia; hematoxylin–eosin).

Sample 2: one wound sample of each animal was stored at −70 °C for biochemical analysis (hydroxyproline and MPO assays), and the other wound sample was used to perform microbiological assay.

### 4.6. CFU Determination from Ulcers

Tissue from Sample 2 (aforementioned) was thawed and packed into 50-milliliter falcon tubes (BD). Further, 3 mL of PBS was added and vortexed under constant stirring for 1 min. After this step, the contents were submitted to serial dilution and plated in a culture medium specific for each microorganism, Candida chromagar for *C. albicans,* and Mannitol salt agar for *S. aureus*, both media from Becton, Dickinson, and Company, Heidelberg, Germany. The plates were incubated at 37 °C for 24 h for *S. aureus,* and 48 h for *C. albicans*. Colonies were counted, and the values were expressed by tissue weight (CFU/mg).

### 4.7. Quantitative Image Evaluation of Inflammatory Infiltrate, Angiogenesis, and Fibroplasia

After euthanasia, tissue samples from the animals of each treatment were stained using the hematoxylin–eosin (HE) staining method. Histological slides were visualized on a LEICA^®^ DM4000B optical microscope with a LEICA^®^ DFC 280 camera with LAS^®^ software, Leica Application Suite for image capture. Then, 4–5 fields per blade were obtained at 200× and 400× magnification. The CellCounter’ plugin of ImageJ 1.48 software was used to count the cells of the inflammatory infiltrate, fibroblasts, and blood vessels in each of the 5 individually quantified images of each animal. Each cell or structure was differentiated by the observer in the image of the lamina increasing by 400×, whereas the blood vessels were quantified by a 200× increase.

This protocol was applied to the animals of each treatment and each time; for the final result, the mean number of inflammatory infiltrates, fibroblasts, and vessels found in the 4–5 fields of each animal was considered. The graphs were plotted with the mean obtained in each group using GraphPad Prism 5 software (GraphPad Software, San Diego, CA, USA), along with the presentation of the standard deviation.

### 4.8. Dosage of Myeloperoxidase Enzyme (MPO)

The neutrophil dosage in the lesion was performed using the myeloperoxidase assay [78]. Tissue samples from Sample 2 were washed with sterile PBS. The supernatant was used for CFU determination, and the washed tissue samples were used for MPO determination. In order to demonstrate that the tissue previously washed with PBS did not affect the MPO measurement, validation assays were performed (data not shown). Biopsies were thawed and buffered in 0.1 M NaCl (Sigma-Aldrich), 0.02 M NaPO 4 (Sigma-Aldrich), 0.015 M NaEDTA (Sigma-Aldrich), and pH 4.7 at –70 °C. Afterwards, buffered tissue samples were triturated in POLYTRON PT 3100 (Avantor) at 13,000 rpm, centrifuged for 15 min at 5000× *g*, and resuspended in 0.05 M NaPO4 (Sigma-Aldrich) buffer (pH 5.4) containing 0.5% hexadecyltrimethylammonium bromide (HTAB) (Sigma-Aldrich). A total of 25 μL of the supernatant and 25 μL of TMB (3, 3′, 5, 5′-tetramethylbenzidine) (Sigma-Aldrich) were added to each well of a 96-well plate. Subsequently, 100 μL of 0.5 mM H2O2 was added to each well, and the reaction was quenched with 4 M sulfuric acid. MPO activity was measured in a plate reader (SpectraMax^®^ 190 Absorbance Plate Reader, Molecular Devices) at 450 nm. Results are reported as the total number of neutrophils/mg tissue by comparing the absorbance of the tissue supernatant to a standard curve.

### 4.9. Evaluation of Collagen Levels

Collagen was assessed by the measurement of hydroxyproline in tissue, following the methodology described by Reddy and Enwemeka (1996) [79], and Dwivedi et al. (2017) with modifications. Briefly, tissue samples frozen at −70 °C (Sample 2) were thawed and incubated for 15 h in an oven at 60 °C in uncapped microtubes. Samples were weighed and homogenized in POLYTRON^®^ PT 3100 at 13,000 rpm with 6N hydrochloric acid (HCl) and transferred to capped glass test tubes. HCL ¨N was added in a proportion of 100 μL per 1 mg of dry tissue, homogenized and subjected to acid hydrolysis, thus incubating the samples in HCL (Sigma-Aldrich) at 130 °C for 4 h.

After incubation, the pH of the samples was adjusted to 7.0 with 2M sodium hydroxide (NaOH) (Sigma-Aldrich). Subsequently, 10 μL of each sample along with 90 μL of 0.056 M T-Chloramine solution were pipetted into 96-well plates incubated in the dark for 25 min. Afterwards, 100 μL of Ehrlich reagent [80] was added for the formation of the chromophore. Thus, the plates were incubated at 60 °C for 20 min as by performing the absorbance reading at 550 nm.

### 4.10. Statistical Analyses

GraphPad Prism software (GraphPad Software, Inc., La Jolla, USA) was employed for statistical analysis with a confidence level of 95%. Parametric statistical analyses were conducted with a one-way ANOVA followed by Bonferroni test.

## 5. Conclusions

The presented results showed that silver nanoparticle formulations (CS and GS), regardless of the route of synthesis being chemical or green, have considerable potential for the treatment of wounds in a diabetic model, especially promoting significant improvement in terms of healing and wound closure after 7 days of treatment when compared to commercial silver sulfadiazine. For both silver nanoparticle types, the mechanism of action was focused on re-epithelization stimulation. For PS formulation, wound closure was mediated by a proinflammatory stimulus, with similar behavior to that of other natural compounds evaluated in the same model. Considering the reproducibility, low cost, and ecofriendliness of green synthesis, besides other properties that it could present, and the same benefits for the pomegranate spray, the produced formulations must be explored more deeply as potential antimicrobial and wound healing agents.

## Figures and Tables

**Figure 1 antibiotics-10-01343-f001:**
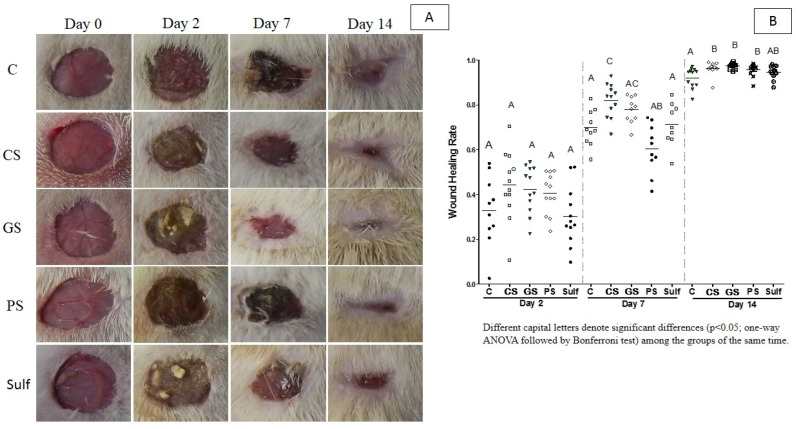
(**A**) Images and (**B**) wound healing rate of different treatments for 14 days. C: control (without treatment); CS: chemical silver nanoparticle spray; GS: green silver nanoparticles spray; PS: pomegranate peel extract spray; Sulf: silver sulfadiazine. Different capital letters denote significant differences (*p* < 0.05; one-way ANOVA followed by Bonferroni test) among the groups of the same time.

**Figure 2 antibiotics-10-01343-f002:**
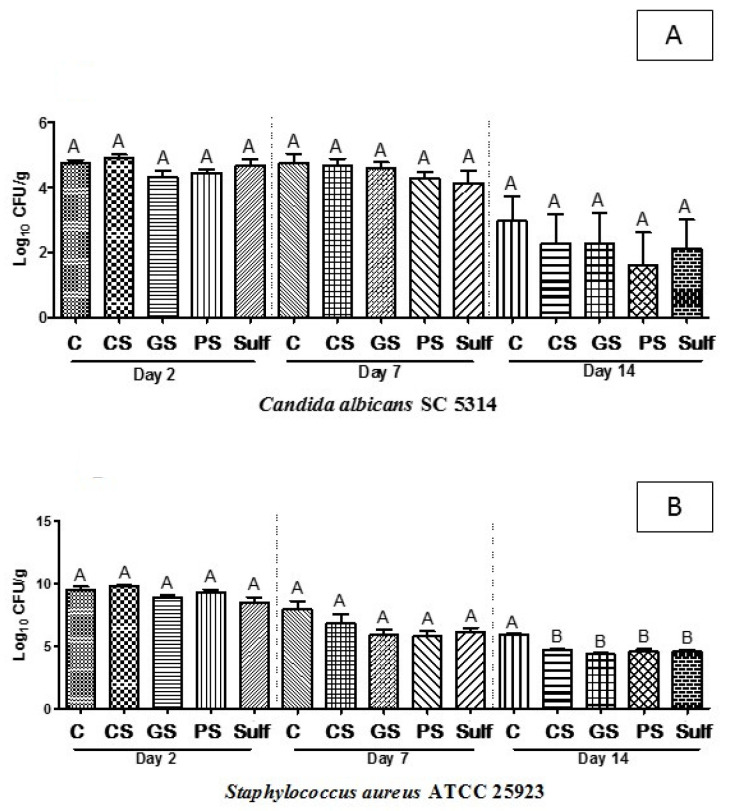
Mean values of logarithm of colony forming units per gram (log10 CFU/g) for mixed (**A**) *C. albicans* and (**B**) *S. aureus* biofilms treated for 14 days. C: control (without treatment); CS: chemical silver nanoparticle spray; GS: green silver nanoparticle spray; PS: pomegranate peel extract spray; Sulf: silver sulfadiazine. Different capital letters denote significant differences (*p* < 0.05; one-way ANOVA followed by Bonferroni test) among the groups of the same time.

**Figure 3 antibiotics-10-01343-f003:**
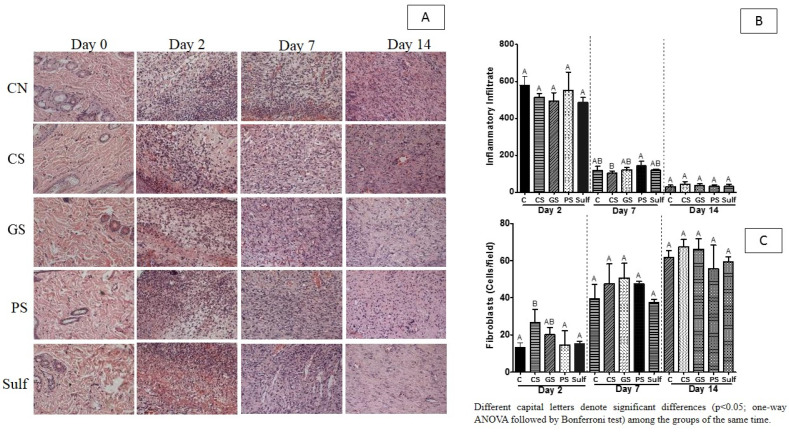
(**A**) histological images of the wounds in different periods of treatment; and fibroblast cells for 14 days of treatment; (**B**) Inflammatory infiltrate in the wounds during 14 days; (**C**) Fibroblasts present in the wounds during 14 days. C: control (without treatment); CS: chemical silver nanoparticle spray; GS: green silver nanoparticle spray; PS: pomegranate peel extract spray; Sulf: silver sulfadiazine. Different capital letters in figures (**B**,**C**) denote significant differences (*p* < 0.05; one-way ANOVA followed by Bonferroni test) among the groups of the same period of time.

**Figure 4 antibiotics-10-01343-f004:**
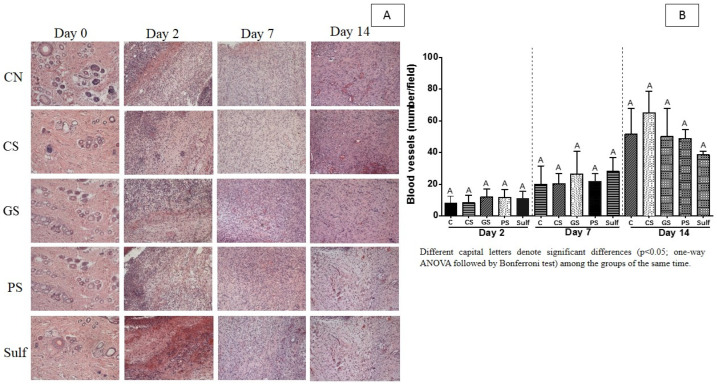
Results of blood vessel count for 14 days of treatment: (**A**) histological images, and (**B**) number of blood vessels. C: control (without treatment); CS: chemical silver nanoparticle spray; GS: green silver nanoparticle spray; PS: pomegranate peel extract spray; Sulf: silver sulfadiazine. Different capital letters in figure (**B**) denote significant differences (*p* < 0.05; one-way ANOVA followed by Bonferroni test) among the groups of the same time.

**Figure 5 antibiotics-10-01343-f005:**
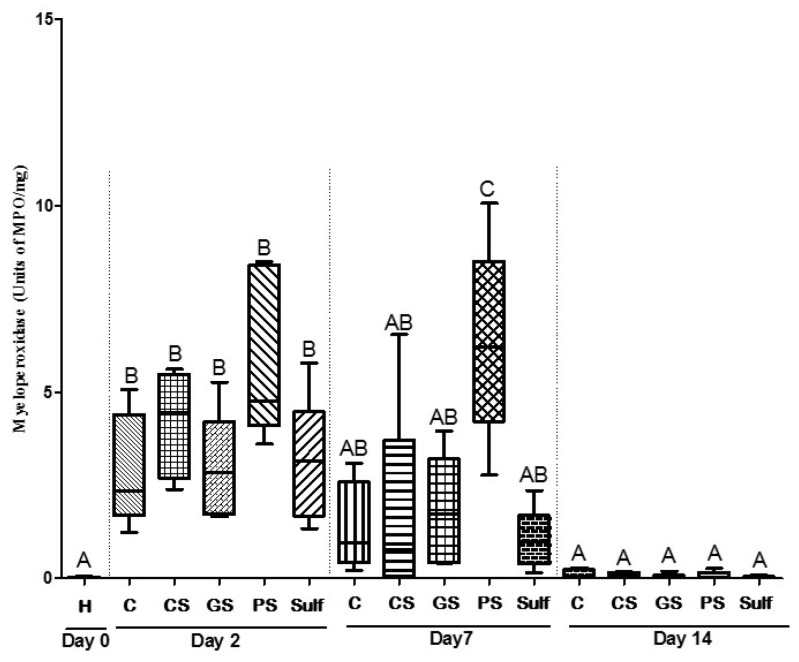
Myeloperoxidase results for 14 days of treatment. C: control (without treatment); CS: chemical silver nanoparticle spray; GS: green silver nanoparticle spray; PS: pomegranate peel extract spray; Sulf: silver sulfadiazine. Different capital letters denote significant differences (*p* < 0.05; one-way ANOVA followed by Bonferroni test) among the groups of the same time.

**Figure 6 antibiotics-10-01343-f006:**
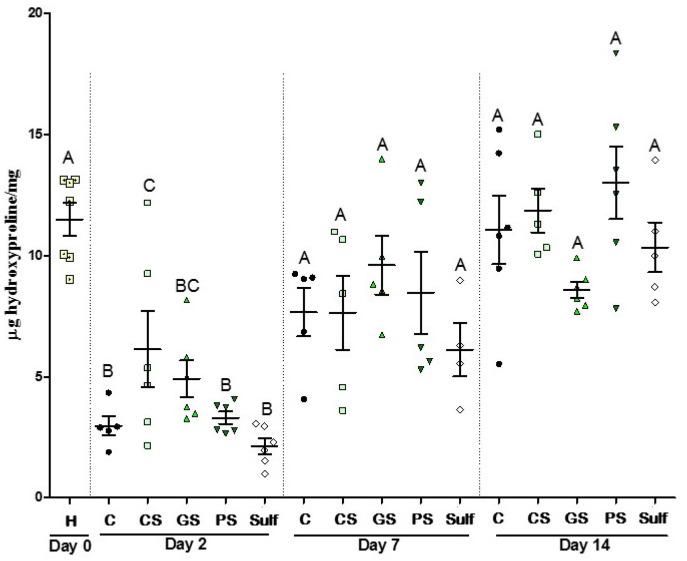
Results of hydroxyproline for 14 days of treatment. C: control (without treatment); CS: chemical silver nanoparticle spray; GS: green silver nanoparticle spray; PS: pomegranate peel extract spray; Sulf: silver sulfadiazine. Different capital letters denote significant differences (*p* < 0.05; one-way ANOVA followed by Bonferroni test) among the groups of the same time.

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
