# Peer review of "Green and Chemical Silver Nanoparticles and Pomegranate Formulations to Heal Infected Wounds in Diabetic Rats"

_antibiotics, 2021, doi:10.3390/antibiotics10111343_

Round 1

Reviewer 1 Report

The subject of the reviewed manuscript concerns the use of preparations containing silver nanoparticles as a potential wound healing agent and antimicrobial drug. The topic is significant from the practical point of view; however, the research on this issue has been conducted for several decades. 
Considering the earlier reports of these authors, the purpose of this work were not specify. Neither has it been specified what the novelty of this paper is. The characteristics (structural, morphological, physicochemical) of silver nanoparticles (produced by chemical (CS) and green silver (GS) methods), which were used in experiments carried out, have been omitted. I propose discussing the obtained results considering the various antimicrobial and anti-inflammatory mechanisms postulated for CS and GS.

Overall remarks:
1. The quality of Figures 1, 2, 3, 4, 6 is bad, the charts are unreadable.
2. Phrases such as "in vivo" should be written Italic.

In my opinion, the publication of the reviewed manuscript in Antibiotics shoud be reconsidered after its  major revision. 

Reviewer 2 Report

In the manuscript, the authors synthesized silver nanoparticles for wound healing and antimicrobial applications. The results indicate that CS present better anti-inflammatory results, and GS and CS have the highest number of fibroblasts. Despite the techniques’ limitations, GS and CS demonstrated considerable potential for managing infected wounds, especially considering no early strategies prior to the drugs, such as the debridement of these wounds, are included.

  1. In the title, green and chemical silver nanoparticles and pomegranate formulations as potential wound-healing and antimicrobial drug, what does it mean GS, CS, and PM formulations as a potential drug?
  2. The authors synthesized silver nanoparticles. However, it was not found the range of particle size and SEM images of particles.
  3. Firstly, this paper studied the antimicrobial activity and wound healing of green and chemical silver nanoparticles and pomegranate formulations, but the significance of these formulations was not put forward in the introduction section. Besides, the innovation of this study was not described in this section, and the reader had to found out it by reading the full text.
  4. Finally, I suggest including the reference “Antimicrobial activity of silver containing crosslinked poly (acrylic acid) fibers Micromachines. 2019;10(12):829 in the introduction section (Line 74).
  5. For Figure 3d, please change the range of the energy axis for easy view.
  6. Some sentences are very long to follow.

The presented results showed that silver-nanoparticle formulations (CS and GS), regardless of the route of synthesis being chemical or green, have considerable potential 495 for the treatment of wounds in a diabetic model, especially promoting significant improvement in terms of healing and wound closure after 7 days of treatment when compared to commercial silver sulfadiazine.

And

Considering the reproducibility, low cost, and eco-friendliness of green synthesis, besides other properties that it could present, and the same benefits for the pomegranate spray, the produced formulations must be explored more deeply as potential antimicrobial and wound-healing agents.

And

Indeed, in view of the potential application of AgNPs for sundry purposes in different fields such as electronics, catalysis, and optics, summer items, and in several biomedical applications, especially due to their broad-spectrum antimicrobial activity [20], the increase in AgNP production per year, according to Rónavári et al (2017), may, globally, be near to hundreds of tons.

Some English typos and styles need to be addressed along with the manuscript. Please check this. patients at risk are diabetic people (high-risk patients with diabetes), due (to) their versatile functions such as antimicrobial, and GS and CS have the highest number of fibroblasts, etc…

  1. There is A, B, AB, AC, and C in figures (1, 2, 3, 4, and 5). The authors should identify them in each figure. It is confusing to find what is the outcome of these alphabets in the figures. It is the information of statistical analysis in the figures, entitled “Different capital letters denote significant ….” They should be added to the captions of figures.

  1. The authors compared the silver nanoparticles (GS and CS) and pomegranate spray (PS) with the control group (C) and sulfadiazine treatment (Sulf). However, it is important to which one of the spraying methods is better compared with other methods and why?

  1. Unit and dimension should be added to figure 3 b).

  1. Line around A, B, and C in the figures should be deleted.

  1. The formulations containing silver nanoparticles (SC and SG) presented the best results when compared to the pomegranate-spray formulation without nanotechnology (SP). What does without nanotechnology mean here?

  1. In the discussion section, there is many references from other findings. The discussion section typically includes the following components: (a) the significance of the study, (b) interpretations of the significant results, (c) implications, (d) limitations, and(e) future studies. The reviewer suggests summarizing this section and discusses the findings of this paper. Other sentences should be removed or move to the introduction section.
  2. The method section in line 382 should include all the procedures the synthesis and spray preparation. The authors need to provide the detail of the process.

  1. The “4. Materials and Methods” section should be before “result in section.” The material section should also endorse vendors of NaCl, NaPO4, NaEDTA, POLYTRON, hexadecyltrimethylammonium bromide (HTAB), TMB (3, 3 ', 5, 5'-tetramethylbenzidine), and H2O2.

  1. In the Abstract, title, and introduction sections, the authors think “Green Chemistry”. But NaCl, NaOH, and some materials can be seen during the material synthesis and wound healing treatments. Why do authors think these methods are green. Authors should find some evidence for the green chemistry of silver nanoparticles in the literature to approve their hypothesis.

Reviewer 3 Report

Comments

Title: Green and chemical silver nanoparticles and pomegranate formulations as potential wound-healing and antimicrobial drug

The work presented in this article does not meet sufficient data of interest to the readers of Antibiotics. It represents a relatively in-vivo study comparing green and chemical silver nanoparticles. The silver nanoparticles have been evaluated in rat models. However, there is no significant difference between green synthesis and chemical synthesis of silver nanoparticles at least in figure 1 B.

If the authors want to show there is no difference between Green and chemical silver nanoparticles, it is recommended to rewrite their manuscript and resubmit again.

Reviewer 4 Report

This is a very well-written manuscript that unfortunately suffers from a major lack of significant results. A variety of appropriate techniques were used and applied well to study wound healing in mice treated with a variety of nanoparticles. The results showed that no significant differences occurred between control(C), no treatment, and tests of the various formulations for monitoring important parameters in wound healing. 

Despite their extensive work, no significant results were obtained and I cannot recomend publication. 

Round 2

Reviewer 1 Report

I propose to accept this paper in the present form. However, the novelty aspects of this article are weak.

Author Response

Point1. I propose to accept this paper in the present form. However, the novelty aspects of this article are weak.

Response: We really thank the reviewer for accepting our paper in the present form. As we wrote at the end of the introduction section our study was developed with researchers from multidisciplinary areas, with the mutual goal of developing a spray containing silver nanoparticles synthesized through pomegranate peel extract to treat infected wounds in diabetic patients and people with pressure ulcer. The choice of using pomegranate peel extract is due to its great phar-macological properties including anti-inflammatory, antioxidant and antimicrobial po-tential. The fact that the present study was developed in partnership with the industry gave us a very optimistic perspective regarding the commercialization of our idea.

Reviewer 3 Report

Thanks for the modification, but unfortunately, the changes are not sufficient to get acceptance. Please redesign all experimental and result parts of the research if the authors need to highlight one of the substances.

Author Response

Thanks for the modification, but unfortunately, the changes are not sufficient to get acceptance. Please redesign all experimental and result parts of the research if the authors need to highlight one of the substances.

Response: Respectfully, redesign all experimental and result parts of our study is not the point since our purpose was not only to highlight one of the sprays prepared. 

Reviewer 4 Report

Please, explain the statistical significance of the results.

For example, on Fig 1 B, provide detailed statistical significance of CS, GS, PS, Sulf bars as compared to the control C. What is the meaning of A, C, AC, AB, A on top of the bars on Fig 1 B?

Wound healing rates look the same as compared to the control.

Clarify also results for which no diferences in capital letters occur on Fig 3 B and C. This certainly means no statistical significance of the results as compared with the control.

On Figure 4, different capital letters do not occur meaning no difference between treated and non treated animals. The same occurs ondays 2 and 14 on Figure 5.

Please, give a better general discussion on the statistical significance of the results (if any). 

Author Response

Please, explain the statistical significance of the results.

Point 1: For example, on Fig 1 B, provide detailed statistical significance of CS, GS, PS, Sulf bars as compared to the control C. What is the meaning of A, C, AC, AB, A on top of the bars on Fig 1 B?

Response: As it was shown in the legend of figure 1 B the statistical significance among the groups is indicated by differente capital letters in the same period of the time evaluated. The meaning of A, C, AC, AB, A was explained before.  

Point 2: Wound healing rates look the same as compared to the control.

Response: Wound healing rates shown in the photographs are qualitative data, which by themselves will not prove what is/are the best treatment to heal the infected wound.  

Point 3: Clarify also results for which no diferences in capital letters occur on Fig 3 B and C. This certainly means no statistical significance of the results as compared with the control.

Response: CS spray was different from the control at day 7 in figure 3B,  as well as at day 2 in figure 3C.

Point 4: On Figure 4, different capital letters do not occur meaning no difference between treated and non treated animals. The same occurs ondays 2 and 14 on Figure 5.

Response: For those parameters there were no statistically significant differences compared to the control group. However, by no means do those results demerit our entire study. 

Point 5: Please, give a better general discussion on the statistical significance of the results (if any). 

Response: It had already been done based not only on the statistical significance found.